# Proteomic Analysis of Salivary Extracellular Vesicles from COVID-19 Patients Reveals a Specific Anti-COVID-19 Response Protein Signature

**DOI:** 10.3390/ijms25073704

**Published:** 2024-03-26

**Authors:** Laura Weber, Alfredo Torres, Ornella Realini, María José Bendek, María Luisa Mizgier, Claudia Brizuela, David Herrera, Fermín E. González, Alejandra Chaparro

**Affiliations:** 1Department of Pathology and Conservative Dentistry, Faculty of Dentistry, Universidad de los Andes, Santiago 7620060, Chile; lweberaracena@gmail.com (L.W.); mbendek@miuandes.cl (M.J.B.); mlmizgier@gmail.com (M.L.M.); cbrizuela@uandes.cl (C.B.); 2Laboratory of Experimental Immunology & Cancer, Faculty of Dentistry, Universidad de Chile, Santiago 8380492, Chile; atorres@odontologia.uchile.cl; 3Department of Conservative Dentistry, Faculty of Dentistry, Universidad de Chile, Santiago 8380492, Chile; 4Centre for Biomedical Research and Innovation (CIIB), Periodontal Research Laboratory, Universidad de los Andes, Santiago 7620060, Chile; or.realini@gmail.com; 5Department of Periodontology, Faculty of Dentistry, Universidad Complutense de Madrid, 28040 Madrid, Spain; davidher@odon.ucm.es

**Keywords:** extracellular vesicles, saliva, COVID-19, proteomics

## Abstract

Despite the understanding of the coronavirus disease-19 (COVID-19), the role of salivary extracellular vesicles (sEVs) in COVID-19 remains unclear. Exploring the proteomic cargo of sEVs could prove valuable for diagnostic and prognostic purposes in assessing COVID-19. The proteomic cargo of sEVs from COVID-19(+) subjects and their healthy close contacts (HCC) was explored. sEVs were isolated by ultracentrifugation from unstimulated saliva samples, and subsequently characterized through nanoparticle tracking, transmission electron microscopy, and Western blot analyses. The proteomic cargo of sEVs was processed by LC-MS/MS. sEVs were morphologically compatible with EVs, with the presence of Syntenin-1 and CD81 EV markers. The sEV pellet showed 1417 proteins: 1288 in COVID-19(+) cases and 1382 in HCC. In total, 124 proteins were differentially expressed in sEVs from COVID-19(+) subjects. “Coronavirus-disease response”, “complement and coagulation cascades”, and “PMN extracellular trap formation” were the most enriched KEGG pathways in COVID-19(+) cases. The most represented biological processes were “Hemoglobin and haptoglobin binding” and “oxygen carrier activity”, and the best-denoted molecular functions were “regulated exocytosis and secretion” and “leucocyte and PMN mediated immunity”. sEV proteomic cargo in COVID-19(+) suggests activity related to immune response processes, oxygen transport, and antioxidant mechanisms. In contrast, in HCC, sEV signature profiles are mainly associated with epithelial homeostasis.

## 1. Introduction

The infection caused by the severe acute respiratory syndrome Coronavirus-2 (SARS-CoV-2), known as coronavirus disease 2019 (COVID-19), was first reported in December 2019 and has since led to a global pandemic (WHO Coronavirus). Severe cases of COVID-19 are characterized by immune dysregulation, increased release of pro-inflammatory cytokines, and the development of respiratory distress syndrome, sepsis, thromboembolism, and multiorgan failure [1,2,3,4]. Various transmission routes of SARS-CoV-2, including direct and contact transmission, have been proposed [5]. Recent studies have highlighted saliva as a potential reservoir for COVID-19, evidenced by the successful detection of SARS-CoV-2 RNA in saliva samples [6,7,8]. Moreover, the oral cavity may serve as an early infection site, where SARS-CoV-2 can directly infect and replicate in the oral mucosa or salivary glands, potentially playing a crucial role in transmitting the virus to the lungs or gastrointestinal tract through saliva ingestion [9].

While numerous biomarkers for COVID-19 diagnosis have been extensively investigated, the focus has predominantly been on blood-based molecular markers [10]. In contrast, saliva offers a compelling alternative with significant advantages, including non-invasiveness, self-sampling, and cost efficiency [11,12]. Whole saliva encompasses a diverse range of molecules, such as proteins, peptides, DNAs, RNAs, hormones, cytokines, enzymes, antibodies, and metabolites, originating from major (parotids, submandibular, and sublingual) and minor salivary glands [13,14]. The composition of saliva is influenced by natural circadian rhythms, mastication, dietary components, oral and systemic diseases, and amylase activity, among other factors [15,16]. Due to their proximity to blood vessels, salivary glands serve as a rich source of metabolite exchange between the oral cavity and the systemic circulation [17]. The detection of molecules found in human serum in saliva suggests that saliva could act as a proxy for disease-related circulating plasma biomarkers, and the emerging field of salivary proteomics holds promise for future diagnostic applications [14].

Recent analyses of the salivary proteome and transcriptome have underscored the diagnostic potential of saliva [12]. Extracellular vesicles (EVs) are nanoparticles secreted by various cell types in different biological fluids, including saliva, containing several proteins and RNAs that could serve as biomarkers [18]. EVs play roles in distant cell communication, protein and RNA release mechanisms, expulsion of obsolete cell and membrane constituents, and antigen presentation, among other functions [19]. Salivary extracellular vesicles (sEVs) are gaining attention as functional carriers and potential biomarkers for various diseases, displaying a heterogeneous population reflective of the origin of different salivary glands [20]. While the biological functions of sEVs in COVID-19 remain unclear, several studies suggest that sEVs could be relevant biomarker candidates for this disease [21,22]. This report aims to isolate and characterize sEVs from COVID-19-positive (+) subjects and their in-house negative healthy close contacts (HCCs) and explore their proteomic cargo using a mass-spectrometry-based proteomics approach.

## 2. Results

### 2.1. Demographic and Clinical Variables

A total of 21 subjects participated in this study, consisting of 11 (52.38%) individuals with symptomatic COVID-19(+) and 10 (47.62%) in-house negative and healthy close contacts (HCCs), confirmed by real-time quantitative polymerase chain reaction (RT-qPCR) of nasopharyngeal swabs. The median age was 40 years (interquartile range, IQR = 25), with a distribution of 42.86% females and 57.14% males. Smokers accounted for 27.3% of the COVID-19(+) subjects and 20% of the HCCs. Notably, 9.1% of the COVID-19(+) patients presented with related comorbidities, while 40% of the HCCs had comorbidities (Table 1). Saliva samples were taken from all the recruited population.

### 2.2. sEV Characterization

There was no difference in the total sEV concentration in the COVID-19(+) subjects compared to the HCCs (3.37 × 10^9^ particles/mL, IQR = 5.35 × 10^9^ vs. 2.02 × 10^9^ particles/mL, IQR = 7.03 × 10^8^ particles/mL, respectively; *p* = 0.47) (Figure 1A,B). In addition, there were no differences in the subpopulation analysis of small sEVs (1.73 × 10^9^ particles/mL, IQR = 2.94 × 10^9^ vs. 6.11 × 10^8^ particles/mL, IQR = 8.86 × 10^8^ particles/mL, respectively; *p* = 0.50) and large sEVs (1.64 × 10^9^ particles/mL, IQR = 2.87 × 10^9^, 1.42 × 10^9^ particles/mL, IQR = 3.10 × 10^8^ particles/mL respectively; *p* = 0.48) (Figure 1B). Furthermore, the size distribution (mode) and the ratio of small/large sEVs were not significantly different between the groups (Figure 1C,D). However, the evaluation of the positive markers by Western blot (WB) analysis showed the presence of tetraspanin CD81 and the endosome marker Syntenin-1, and a minimal presence of albumin contaminant in the salivary sEV-enriched pellet (Figure 1E).

In addition to these results, the transmission electron microscope (TEM) analysis confirmed the presence of nanoparticles compatible with the size and characteristic morphology of EVs, showing a round-shaped structure surrounded by a bi-layered membrane (Figure 1F).

### 2.3. Proteome Identification

To evaluate changes in the sEV proteome induced by symptomatic COVID-19 infection, proteomic data were visualized using a Venn diagram and a volcano plot, comparing the protein profiles of the COVID-19(+) and HCC groups (Figure 2A,B). A total of 1417 proteins were identified, with 1382 and 1288 proteins found in the HCC and COVID-19(+) groups, respectively (Figure 2A). A qualitative and quantitative profile was distinctive in each group: 35 proteins were exclusively found in the COVID-19(+) samples, while 129 were found in the HCC samples (Figure 2A). In terms of quantitative analysis, 161 proteins exhibited significant differences in their abundance (≥2-fold change, *p* < 0.05) (Figure 2B), with 89 proteins being more abundant in the COVID-19(+) group and 72 in the HCC group. The comprehensive proteome from sEV pools from COVID-19(+) and HCC subjects is detailed in Appendix A.

### 2.4. Functional Enrichment Analysis of Differentially Expressed Proteins

To elucidate the biological functions represented in the proteomic profiles observed in sEVs from COVID-19(+) and HCC groups, a bioinformatic enrichment analysis based on the KEGG and GO classification systems was performed.

In the KEGG database analysis, the COVID-19(+) group exhibited enrichment in immune pathways, such as “PMN extracellular trap formation”, “Complement and coagulation cascades”, and “Coronavirus disease”. Notably, “Coronavirus disease” emerged as the most represented term based on the number of associated proteins and the *p*-value (Figure 2C). Additionally, “Ferroptosis” was among the prominently represented terms (Figure 2C). In contrast, the HCC group showed enrichment in terms like “Staphylococcus aureus infection”, “Estrogen signaling pathway”, and “Amoebiasis” (Figure 2D).

GO analysis in the category of “Molecular function” for the COVID-19(+) group highlighted enriched functions associated with “Hemoglobin and haptoglobin binding”, “Oxygen carrier activity”, “Antioxidant- and peroxidase-activity”, and “Oxygen binding” (Figure 2E). In the HCC group, the most represented sub-categories were “Structural molecule activity”, “Extracellular structural constituent”, and “Structural constituent of cytoskeleton and epidermis” (Figure 2F). 

Within the GO analysis in the “Biological process” category, the COVID-19(+) group revealed enriched terms associated with “Exocytosis” processes, “Secretion and exporting from the cell”, “Cell activation” and terms linked to immune processes emphasizing “Leukocyte mediated immunity”, “PMN-mediated immunity”, and “PMN activation and degranulation” (Figure 2G). In contrast, the HCC group exhibited proteins linked to “Cornification”, “Keratinization”, “Keratinocyte differentiation”, and “Skin development and epithelial cell differentiation” (Figure 2H).

A hierarchical clustering network summarized the correlation between significant pathways of exclusive and more abundant proteins in the COVID-19(+) and HCC groups (Appendix A), highlighting proximity between terms like “Coronavirus disease”, “PMN extracellular trap formation”, “Complement and coagulation cascades”, and “Staphylococcus aureus infection”. The interactome network of pathway terms also reflected proximity between the enriched terms (Appendix A), with darker nodes indicating more significantly enriched sets of proteins, larger dots representing more significant *p*-values, and thicker edges signifying more overlapping proteins. Appendix A illustrates the enrichment analysis based on the KEGG and GO classification systems, evaluating all exclusive and more abundant proteins from both groups. Remarkably, the enriched terms identified in the individual group analysis (Figure 2) were consistently represented in the comprehensive assessment of differentially expressed proteins, irrespective of their origin. Moreover, the hierarchical cluster analysis demonstrates the clustering of enriched terms, emphasizing that enriched terms closely associate with each other according to the clinical status represented as COVID-19(+) or HCC.

### 2.5. Identification of Hub Proteins in the Key Modules

To further investigate the functions and mechanisms of the differentially expressed proteins, an identification and characterization of the hub proteins was carried out. Exclusive and more abundant proteins from both the COVID-19(+) and HCC groups were uploaded into the STRING online database for protein–protein interaction analysis, and the resultant networks were visualized in the Cytoscape environment. Utilizing the CytoHubba plugin with the Maximal Clique Centrality (MCC) method, the top 10 hub proteins in both groups were determined (Figure 3).

In the COVID-19(+) group, the identified hub proteins included FN1, FGA, TF, HP, CLU, FGB, FGG, ALB, CP, and C3 (Figure 2B and Figure 3B). Conversely, in the HCC group, HSPG2, COL4A1, COL4A2, COL1A1, COL6A2, LAMA5, LAMB1, LAMB2, FBN1, and KRT6A were recognized as hub proteins (Figure 3C,D). Significantly, the hub proteins of the COVID-19(+) group aligned with the most enriched terms of the total set of exclusive and more abundant proteins previously identified in this group (Figure 3E,F). The detailed interaction of functional connections between the top 10 hub proteins from the HCC and COVID-19(+) groups, along with their assigned Gene Ontology (GO) terms, is depicted in Figure 4. Additionally, the currently known functions of the hub proteins of the COVID-19(+) and HCC groups are described in Table 2.

## 3. Discussion

EVs constitute heterogeneous populations of membrane-derived vesicles, released by different cell types, including epithelial, endothelial, immune, neuronal, and platelet cells [23]. EVs play diverse and crucial roles in physiological and pathological cellular processes, providing insights into the disease state of the originating cells [23,24,25]. Additionally, EVs serve as carriers of signals, facilitating the transfer of proteins and nucleic acids from donor cells to target cells, either locally or at a distance. Consequently, they have the potential to influence the function and phenotype of the target cells [26,27].

Notable contributions have been made to the understanding of COVID-19 through proteomics [28,29,30,31], and several studies have explored the saliva proteome during SARS-CoV-2 infection [32,33,34]. Nevertheless, to our knowledge, this study marks the first attempt to assess extracellular EVs in saliva from COVID-19 patients, comparing them to their healthy “in-house” close contacts.

The differences in the clinical characteristics of the participants comprising the COVID-19(+) and the HCC groups were not significant in gender or smoking; however, differences were significant when assessing co-morbidities. Although the literature suggests no differences in EV concentration or size with age and sex, evidence suggests that EV cargo is influenced by health status and other systemic factors [35]. To address the challenges arising from heterogeneity and individual variations in this analysis, saliva samples from each patient were combined to create a more representative sample set. 

We successfully isolated salivary EVs from COVID-19(+) subjects, evaluating their concentration, size distribution, morphology, and the presence of characteristic surface and cytosolic markers. Our findings showed that the EVs exhibited a round-shaped structure surrounded by a bi-layered membrane, consistent with the typical morphology of EVs characterized in other biological fluids [23,36,37,38,39]. Additionally, the size distribution of EVs in saliva revealed the presence of particles within the recognized range for small EVs (<200 nm) and large EVs (>200 nm) [23]. Although both, large and small EVs were increased in the COVID-19(+) group compared to the HCC group, no significant differences in EV concentration were observed. However, qualitative and quantitative disparities in the proteomic cargo were identified.

Given that EVs encompass markers of the immune–inflammatory response derived both locally and systemically, they represent a potential foundation for site-specific biomarker assessments for systemic diseases in oral fluids [40,41,42]. Additionally, the diversity of EVs cargo, offering a multitude of diagnostic parameters, has the potential to enhance both sensitivity and specificity in diagnostics and the non-invasive nature of the collection of oral fluids makes the characterization and quantification of EVs, particularly in saliva, highly advantageous. This approach holds promise in the monitoring of severe COVID-19 patients and those who do not show signs of recovery, potentially serving as a reliable assessing tool for predicting at-risk patients. 

In this study, we hypothesized that SARS-CoV-2 infection induces characteristic changes in the proteome that can be detected in the sEVs of COVID-19(+) patients. The Gene Ontology (GO) biological process enrichment analysis of differentially expressed proteins in the COVID-19(+) group revealed that “Regulated exocytosis”, “Exocytosis”, “Secretion by cell”, “Export from cell”, and “Secretion” were the top five most enriched terms. This alignment with active processes of exocytosis and cell secretion is consistent with the nature of the analyzed samples derived from sEVs. 

Immune-mediated pathology emerges as a key player in the clinical severity of COVID-19. The complement system and the coagulation cascade form an intricate network, maintaining a delicate balance between the two pathways. The COVID-19 pandemic has highlighted the strong link between hemostasis and the immune response [43]. The cytokine storm has been implicated in downstream intravascular coagulopathy, characterized by thromboembolism in the autopsies of COVID-19(+) patients [44,45,46]. Recent data further suggest that the complement cascade and neutrophils contribute to the dysregulated immune response leading to hyperinflammation and thrombotic microangiopathy in COVID-19 infection [47,48,49,50,51,52]. Reactive oxygen species (ROS) formed during COVID-19 infection directly activate neutrophil extracellular traps (NETs). Excessive NET generation is associated with the development of tissue damage during acute lung injury and acute respiratory distress syndrome, owing to the formation of inflammatory cascades in a vicious loop with interleukin (IL)-1β [52,53]. Noteworthily, according to the Kyoto Encyclopedia of Genes and Genomes (KEGG) pathway analysis, the most enriched pathways in the sEVs of the COVID-19(+) group were “Coronavirus disease”, “Complement and coagulation cascades”, “PMN Extracellular trap formation”, and “Pentose phosphate pathway”. Our findings indicate that COVID-19 infection induces characteristic changes in the proteome of sEVs in infected patients, associating this disease with a specific immune host response. This observation aligns with previous studies that have demonstrated the exacerbation of clinical manifestations of COVID-19 due to thrombosis, intravascular coagulation disturbances, cytokine storms [44], and shifts in glycolytic activity [54]. 

Remarkably, symptomatic COVID-19 prompts a metabolic shift toward enhanced glycolytic activity at the cellular level [54]. This shift in glycolysis can impact other metabolic pathways, such as the non-oxidative pentose phosphate pathway (PPP) [55]. The non-oxidative PPP converts glycolytic intermediates into ribose-5-phosphate, essential for nucleic acid synthesis, and sugar phosphate precursors crucial for amino acid synthesis [55]. Interestingly, the inhibition of the non-oxidative PPP disrupts SARS-CoV-2 replication [54], suggesting that the enrichment of the KEGG term “Pentose phosphate pathway” observed in the present study, could be part of a metabolic change driven to support virus replication [54,55,56,57].

The identification of metabolic and molecular alterations further reveals the association of infection with host responses in the pathogenesis of COVID-19. Consequently, additional analyses were conducted to identify the hub proteins associated with COVID-19 pathogenesis. Hub proteins, characterized by high intramodular connectivity, are integral to the network’s topology. In this study, ten hub proteins were defined based on the highest connectivity of exclusive and more abundant proteins in both groups (Figure 3). These proteins were selected through a combined analysis of gene intramodular connectivity and protein–protein interactions using the MCC algorithm in the STRING database and Cytoscape software version 3.10.1. Proteins with the highest MCC score tend to encode essential proteins. These selected hub proteins were further analyzed for their relevance in representing KEGG pathways. Significantly, they mirrored the same terminologies defined by the entire network, providing robust evidence for the reprogramming of the host’s cellular metabolism following SARS-CoV-2 infection. In this study, functional interactions of hub proteins from the sEVs of the COVID-19(+) group were related to immune response, platelet degranulation, the regulation of cell death, and the response to metal ions.

The control group in this study comprised in-house close contacts (the HCC group) who remained uninfected as healthy individuals. The proteomic profile of sEVs from controls revealed proteins associated with molecular functions and biological processes primarily related to homeostatic mechanisms, such as “Structural molecule activity”, “Extracellular constituent of cytoskeleton and epidermis”, “Keratinocyte differentiation”, and “Skin development and epithelial cell differentiation”. This observation gains significance considering that the observed proteomic profile may robustly represent specific differences during COVID-19 infection between the two groups.

The data obtained in this study not only demonstrate the feasibility of isolating sEVs in COVID-19 diseased patients but also contribute to the growing body of knowledge surrounding the potential diagnostic utility of sEVs. The differentiated proteomic cargo profiles observed between the HCC and COVID-19(+) groups underscore the potential of salivary EVs as valuable biomarkers in the ongoing efforts to understand and manage this viral infection. 

While our study offers valuable insights into the proteomic signature of sEVs in COVID-19 patients, we acknowledge certain limitations. The small sample size restricts the generalizability of our findings, emphasizing the need for future research with larger cohorts to validate our results. Moreover, conducting functional validation experiments is essential to confirm the biological relevance of our findings. Nonetheless, our results provide a foundation for future research aimed at further elucidating the functional significance and diagnostic applicability of the identified proteins and their enriched-term annotations.

In summary, this study highlights quantitative and qualitative differences in the proteome of sEVs from COVID-19 patients compared to in-house close contacts as negative controls. The enriched terms for these changes at the proteomic level suggest that the pathogenesis of COVID-19 induces molecular changes and cellular reprogramming that can be identified in sEVs from the saliva of COVID-19 patients. Further research is warranted to assess if these molecular changes at the protein level hold potential for diagnostic and monitoring applications during COVID-19 infection.

## 4. Materials and Methods

### 4.1. Study Design

A cross-sectional study was meticulously designed to investigate the landscape of sEVs in the context of symptomatic COVID-19 infection (Figure 5). COVID-19(+) patients identified through real-time quantitative polymerase chain reaction (RT-qPCR) of nasopharyngeal swabs were purposively chosen from among those diagnosed at the Health Care Centre of the University of the Andes and Davila Clinic in Santiago, Chile. To ensure a comprehensive examination, at least two healthy close contacts (HCC), residing with each COVID-19(+) subject, were also called to participate. Upon confirmation of a positive diagnosis (within 48 h of sampling), both the COVID-19(+) subjects and their HCC were approached and visited at their residences. The control group, comprising in-house healthy close contacts, was defined as individuals with a negative RT-qPCR result sharing a residence with a confirmed COVID-19(+) case. The home visits encompassed an anamnesis, a comprehensive symptoms questionnaire, and the collection of saliva samples. The recorded information included age, gender, body mass index, smoking status, the patient’s history of comorbidities, and potential symptomatology related to COVID-19, all meticulously cataloged in a pre-designed database.

The research protocol obtained ethical approval from the Ethics Committees of both the Davila Clinic and the University of the Andes. The study adhered strictly to the principles outlined in the Helsinki Declaration of 1973, as revised in 2003. Detailed explanations of the study protocol were provided to all participants, and their informed consent was obtained through the voluntary signing of an informed consent form. In the case of minors or individuals below the age of majority, both the parents or guardians and the minors themselves signed an informed assent form, ensuring the comprehensive ethical conduct of the study.

### 4.2. Saliva Samples Collection 

All participants were instructed to provide an early morning, unstimulated saliva sample through self-collection. The collection process involved passive drooling for 1–3 min into a 50 mL sterile conical centrifuge tube (Falcon^®^, Corning, Glendale, AZ, USA). To ensure the integrity of the samples, participants were advised to abstain from eating for two hours and from drinking for 30 min before saliva collection. Furthermore, participants were recommended to refrain from consuming beverages, tobacco, or gum for the 30 min preceding collection.

Participants were specifically requested to collect between three to five ml of saliva, with specimens considered acceptable if a threshold of 1 mL of saliva was obtained. Subsequently, the collected saliva in the sterile tube was refrigerated and transferred to the laboratory within a maximum timeframe of 4 h. Upon arrival at the laboratory, the samples were stored at −80 °C until sample preparation for further analyses.

### 4.3. Separation of sEVs 

To overcome the challenges posed by heterogeneity and individual variations in subsequent analysis, saliva samples from each patient were pooled to obtain a more representative sample set, allowing for the identification of common trends and meaningful biological insights across the sample cohort. sEVs were isolated from saliva samples by ultracentrifugation. Briefly, 2 mL of pooled samples were diluted with filtered PBS 1× (phosphate-buffered saline, containing 137 mM of NaCl, 2.7 mM of KCl, 10 mM of Na_2_HPO_4_, and 1.8 mM of KH_2_PO_4_, pH 7.4). to a volume of 10 mL and centrifuged at 10,000× *g* at 4 °C for 30 min, to remove debris and cell debris. The supernatant was then transferred to a 12 mL ultracentrifuge tube (#3699, Thermo Scientific, Waltham, MA, USA) and centrifuged at 4 °C at 160,000× *g* for 70 min. Subsequently, the supernatant was discarded, and the pellet obtained was resuspended into 10 mL of filtered PBS 1×. The ultracentrifugation was repeated under the same parameters. Upon completion, the supernatant was discarded, and the enriched EV pellets were resuspended into 100 µL of Dulbecco’s phosphate-buffered saline (DPBS). 

### 4.4. Nano-Tracking Particle Analysis 

The concentration (particles/mL) and size distribution (mode) of the sEVs isolated from the saliva samples were analyzed by nano-tracking particle analysis (NTA) using a Nanosight NS300 (Malvern, Worcestershire, UK). All the samples were introduced into the sample chamber and were recorded using a camera level of 10 (slide shutter of 10: slider gain of 73). The NTA post-acquisition settings were optimized and kept constant between samples. Each video was analyzed to give sEV concentration and size distribution.

### 4.5. Western Blot (WB) Analysis 

sEV pellets were prepared with an equal volume of each condition, and negative control of EV-depleted saliva was also included. Protein concentration was measured using the Qubit Protein assay kit (#Q33212, Thermo Scientific, Waltham, MA, USA) according to the manufacturer’s instructions and the Qubit 3.0 Fluorometer (Invitrogen^TM^, Waltham, MA, USA). Later, 10 μg of total proteins were separated by a 12% polyacrylamide gel electrophoresis and then semi-dry transferred onto polyvinylidene difluoride membranes (PVDF; Thermo Scientific, Waltham, MA, USA) for 30 min at 25 V, 1.0 A using a Trans-Blot Turbo transfer system (BioRad, Hercules, CA, USA). The membranes were blocked with 3% bovine serum albumin (BSA) in TBS-Tween (0.1%) for one hour at room temperature under agitation. The blocked membranes were further incubated overnight with the following antibodies: anti-CD81 (SC-166028, Santa Cruz, Santa Cruz, CA, USA), anti-Syntenin-1 (NBP2-76893, Novus Bio, Minneapolis, MN, USA), and anti-Albumin (A3293, Sigma, Kanagawa, Japan). Secondary antibodies, HRP-conjugated anti-rabbit or anti-mouse (LI-COR, Lincoln, NE, USA) were incubated for one hour at room temperature under agitation. Then, membranes were incubated with Super Signal^TM^ West Femto sensitivity substrate (#34094, Thermo Scientific, Waltham, MA, USA) and visualized after appropriate exposition time in a transilluminator G-Box (Sygene, Cambridge, UK). 

### 4.6. Transmission Electron Microscopy (TEM)

The sEV pellets were resuspended in 50 μL of DPBS (Gibco, Billings, MT, USA) and then prepared for microscopy, using 15 μL of the sEV pellets and a 5-fold dilution in double-distilled H_2_O. The samples were incubated for 1 min in ozone pre-treated copper grids, then fixed with Uranyl acetate for 1 min, and then dried in a 60 °C oven for 4 min. After that, the samples were visualized in the microscope, and at least 5 photos per condition were captured. The qualitative morphology assessment of the sEVs was analyzed using the Talos transmission electron microscope (Thermo Scientific, Waltham, MA, USA).

### 4.7. Protein Extraction from sEVs

Protease/phosphatase inhibitor (#1861284, Thermo Scientific, Waltham, MA, USA) was added to each sample at a final concentration of 1×. The samples were then lyophilized and resuspended in 8 M urea with 25 mM ammonium bicarbonate pH 8 and subsequently homogenized using ultrasound for 1 min with 10 s pulses (on/off) at 50% amplitude using a cold bath. Then, they were incubated on ice for 5 min and subsequently centrifuged to remove debris at 20,000× *g* for 10 min at 4 °C. The samples were immediately quantified by Qubit^TM^ 4 (Thermo Scientific, Waltham, MA, USA) using the Qubit Protein Assay kit. A total of 100 μg was taken for tryptic digestion.

### 4.8. Proteolytic Digestion

The resultant proteins were precipitated by adding five volumes of cold acetone and incubated overnight at −80 °C. They were then equilibrated at room temperature for 10 min, centrifuged at 16,000× *g* for 15 min at 4 °C, and the supernatant was discarded. The resulting pellet was washed three times with cold 80% acetone. Subsequently, the protein pellet was allowed to dry in a rotary concentrator. The samples were resuspended in 30 μL of 8 M urea and 25 mM of ammonium bicarbonate. Then, they were reduced with DTT to a final concentration of 20 mM in 25 mM of ammonium bicarbonate and incubated for 1 h at room temperature. Next, they were alkylated by adding Iodoacetamide to a final concentration of 20 mM in 25 mM of ammonium bicarbonate and incubated for 1 h in the dark at room temperature. Subsequently, the samples were diluted 8 times with 25 mM of ammonium bicarbonate. Digestion was performed with trypsin sequencing grade (#V5071, Promega, Madison, WI, USA) in a 1:50 protease/protein ratio (mass/mass) and incubated for 16 h at 37 °C, the digestion reaction was stopped by adding 10% formic acid. The samples were then subjected to Clean Up Thermo Pierce C18 Spin Columns (cat cod 89870, Thermo Scientific, Waltham, MA, USA), according to the supplier’s instructions. Subsequently, the cleaned peptides were dried in a rotary concentrator at 1000 rpm overnight at 10 °C.

### 4.9. Liquid Chromatography–Tandem Mass Spectrometry (LC-MS/MS) Analysis

In total, 200 ng of the peptides obtained in the previous step were injected into a nanoUHPLC nanoElute (Bruker Daltonics, Billerica, MA, USA) coupled to a timsTOF Pro mass spectrometer (“Trapped Ion Mobility Spectrometry—Quadrupole Time of Flight Mass Spectrometer”, Bruker Daltonics) using an Aurora UHPLC column (25 cm × 75 μm ID, 1.6 μm C18, IonOpticks, Melbourne, Australia). Liquid chromatography was performed using a 90 min gradient of 2% to 35% buffer B (0.1% formic acid-acetonitrile). Result collection was performed using TimaControl 2.0 software (Bruker Daltonics) under 10 cycles of PASEF, with a mass range of 100–1700 *m*/*z*, capillary ionization at 1500 V, a capillary temperature of 180 °C, and TOF frequency of 10 KHz at a resolution of 50,000 FWHM. The samples were run in duplicate, and a total of 4 samples were analyzed. To ensure that any contamination or other sources of error were identified, blank runs were added after each analysis of technical replicates to minimize carryover and improve the accuracy of the results.

### 4.10. Protein Identification

The data obtained were analyzed with PEAKS Studio X+ software version 10.6 (Bioinformatics Solutions, Waterloo, ON, Canada) on a data analysis server consisting of 48 cores and 512 Gb of RAM. Mass tolerance parameters of 50 ppm, using monoisotopic masses, and ionic fragments of 0.05 Da were used. Digestion options included trypsin as an enzyme, a specific digestion mode, and a maximum of two missed cleavages per peptide. Carbamidomethylating of cysteine, oxidation of methionine, acetylation of lysine, deamination of asparagine and glutamine, and the carbamylating of lysine and N-terminus were used as post-translational modifications. The database used for identification was the human proteome, available in Uniprot (20,377 entries). False discovery rate (FDR) estimation through a decoy database was included. An FDR ≥ 1% (significance ≥ 20 in PEAKS Studio X+) and 1 minimum unique peptide per protein were used as a filter for identification.

### 4.11. Protein Quantification by Label-Free Quantification

Proteins previously analyzed in the PEAKS Studio X+ search engine were exported in CSV format with the area under the curve values, which were normalized, and subsequently the quantification analyses were performed, using the Positive/Negative ratio. Proteins were represented as a function of the Log2-fold change (Log2FC), using the R package Bioconductor.

### 4.12. Functional Enrichment Analysis of Differentially Expressed Proteins

To attain a better understanding of the biological activities of exclusive and overexpressed proteins in both groups, identification of enriched terms across the Kyoto Encyclopedia of Genes and Genomes (KEGG) pathways, gene ontology (GO) biological process, and molecular function [58] was conducted with the ShinyGO web tool (version 0.77) [59].

### 4.13. Identification of Hub Proteins in the Key Modules

Exclusive and overexpressed protein-coding proteins from both groups were uploaded into the search tool for the retrieval of the interacting proteins (STRING) website (accessed on 9 March 2023 from www.string-db.org) for protein–protein interaction analysis [59], choosing the confidence > 0.4. Cytoscape software (San Diego, CA, USA) was used for the visualization of networks and hub gene selection [60]. The top 10 hub proteins in each group were selected with the maximal clique centrality (MCC) method using CytoHubba version 0.1 [61] plugin software in Cytoscape. According to the MCC ranking, the top proteins were taken as hub proteins and then explored by exploiting database knowledge through UniProt [62].

### 4.14. Data and Statistical Analyses 

The normality of the data distribution was tested using the D’Agostino–Pearson test. The concentration of sEVs was not normally distributed; therefore, a non-parametric test was used. The Mann–Whitney U test was used for comparisons of continuous variables. Data were analyzed using the STATA v16 software (StataCorp, College Station, TX, USA) and GraphPad Prism 8 software. The differences in relative protein abundances between the sample groups were assessed by moderated t-test. Benjamini–Hochberg correction for multiple comparisons was used. Comparisons of relative abundances between the study groups were carried out with a threshold of a ≥2-fold change, according to Levin [63] and differences were considered statistically significant at *p* < 0.05. All analyses were conducted with R software (version 3.5.5).

## 5. Conclusions

Taken together, our data support the notion that the pathogenesis of COVID-19 induces molecular changes and cellular reprogramming that can be identified quantitative and qualitatively in the proteome of sEVs from COVID-19(+) patients compared to in-house close contacts as negative controls. The enriched terms of sEV proteomic cargo in the COVID-19(+) group suggest an association to immune response processes, oxygen transport, and antioxidant mechanisms. In contrast, in the HCC group, enriched terms of sEV signature profiles suggest processes related to epithelial homeostasis.

## Figures and Tables

**Figure 1 ijms-25-03704-f001:**
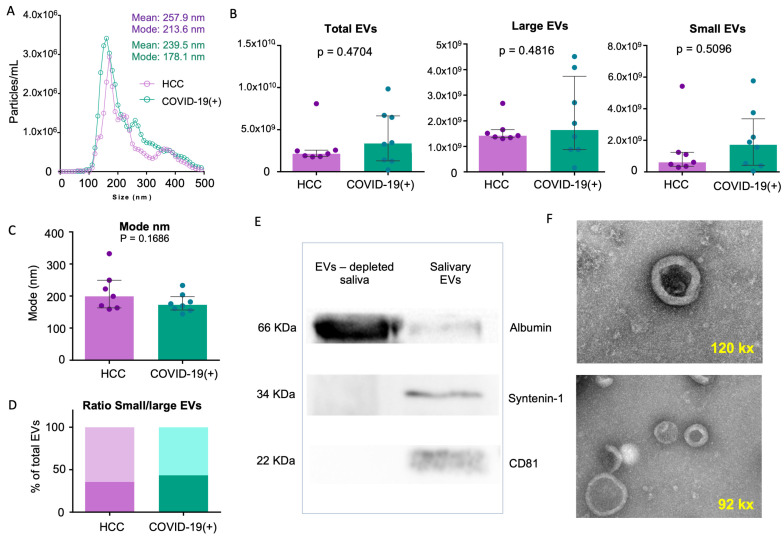
Characterization of sEVs from COVID-19(+) and HCC subjects. (**A**) sEV distribution plot indicating size (nm) and concentration (particles/mL). (**B**) Bar plots for total, large, and small EVs in the HCC and COVID-19(+) groups. (**C**) Mode plot in the HCC and COVID-19(+) groups. (**D**) Size distribution and ratio of large/small plots for sEV subpopulations in the HCC and COVID-19(+) groups. (**E**) EV markers (Syntenin-1 and CD81) detected by WB in salivary EV and EV-depleted saliva. (**F**) Morphology of sEVs by TEM. Abbreviations: sEVs; salivary extracellular vesicles, HCC; healthy close contacts, WB; Western blot, TEM; transmission electron microscope.

**Figure 2 ijms-25-03704-f002:**
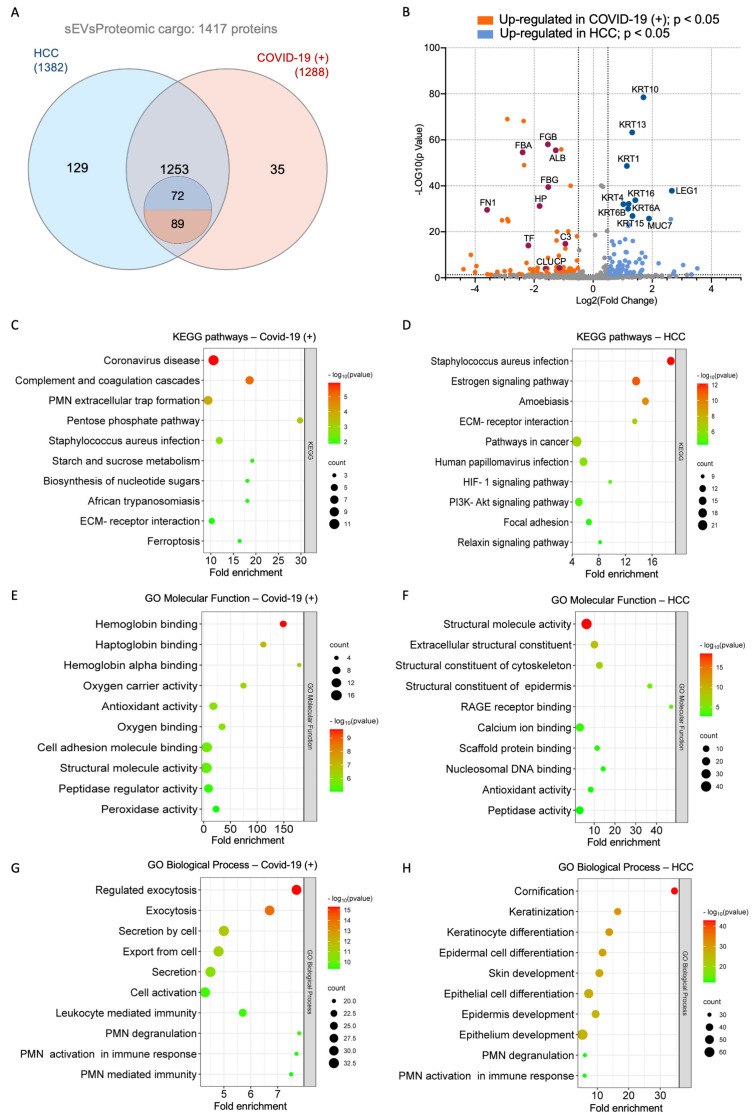
Overall protein profiles of the COVID-19(+) and HCC groups. (**A**) Venn diagram depicting overlapping proteins between the HCC and COVID-19(+) groups, including exclusive and more abundant (differentially expressed) proteins from each group. A total of 1417 proteins were identified, of which 124 were differentially expressed in the COVID-19(+) group. Among these, 35 were exclusive and 89 were more abundant. (**B**) Volcano plot: the log_2_ (fold change) indicates the mean relative abundance for each protein. Each dot represents one single protein. The red and blue areas represent more abundant proteins with significant differences (*p* < 0.05) in the COVID-19(+) and HCC groups, respectively. Dark colored dots represent hub proteins in the COVID-19(+) group. In contrast, the most abundant proteins in the HCC group are represented by dark colored dots, as their hub proteins were mainly exclusively found and were not represented in the plot. (**C**,**E**,**G**) Functional annotations and over-representation analysis of the top 10 most enriched KEGG pathways, GO molecular functions, and GO biological processes, respectively, of the exclusive and differentially expressed proteins (≥2-fold change, *p* < 0.05) in the COVID-19(+) group. (**D**,**F**,**H**). Functional annotations and over-representation analysis of the top 10 most enriched KEGG pathways, GO molecular functions, and GO biological processes, respectively, of the exclusive and differentially expressed proteins (≥2-fold change, *p* < 0.05) in the HCC group.

**Figure 3 ijms-25-03704-f003:**
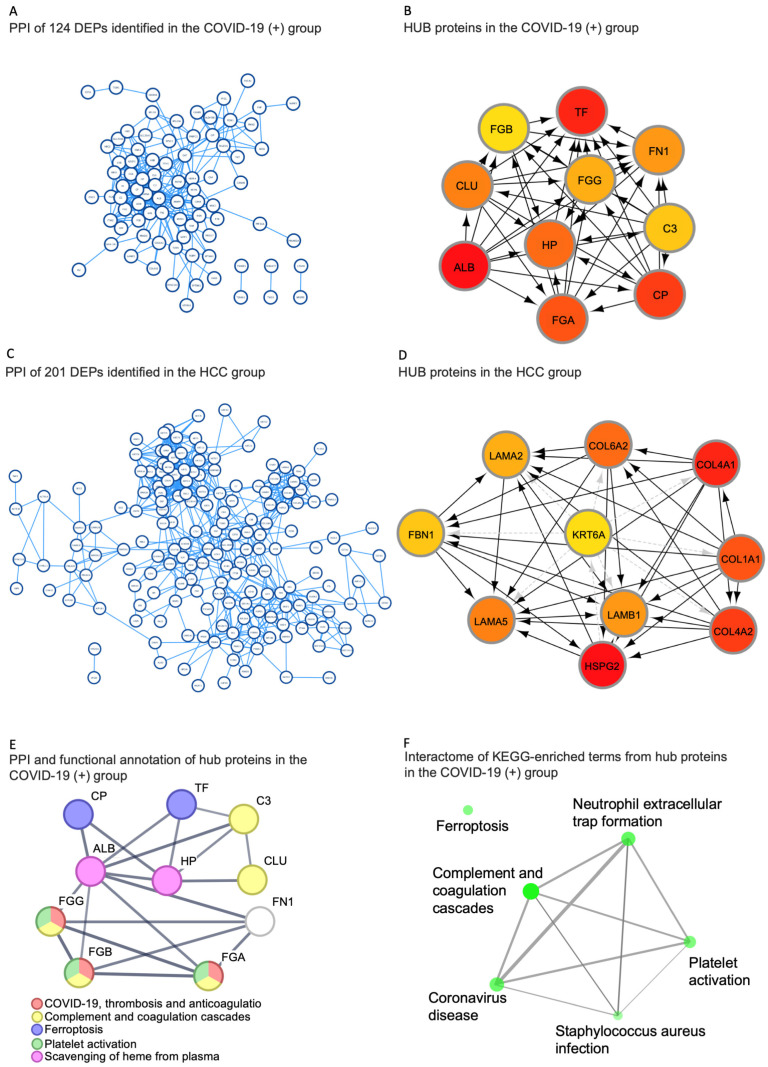
Construction of PPI networks and identification of candidate hub proteins. (**A**,**C**). PPI network analysis in both the COVID-19(+) and HCC groups. In COVID-19(+), the PPI network consisted of 102 nodes and 281 edges, while in HCCs the PPI network consisted of 195 nodes and 610 edges, all of them were visualized in Cytoscape. A comprehensive view of the PPI network is illustrated in Appendix A for both the COVID-19(+) and the HCC groups, respectively. (**B**,**D**) Top ten ranked proteins in network analysis with the MCC method in cytoHubba in COVID-19(+) and HCC, respectively. The top ten nodes are shown with a color scheme from red (highly important) to yellow (very important). (**E**) STRING PPI of hub proteins in the COVID-19(+) group: each color represents an enriched term. (**F**) KEGG pathway terms—the interactome network generated (*p* < 0.005, FDR Q < 0.05) of hub proteins from the COVID-19(+) group: each node represents an enriched pathway term. The nodes are connected by a line whose thickness reflects the percentage of overlapping proteins, and the size of the node corresponds to the number of proteins. Abbreviations: PPI, protein–protein interaction; MCC, maximal clique centrality.

**Figure 4 ijms-25-03704-f004:**
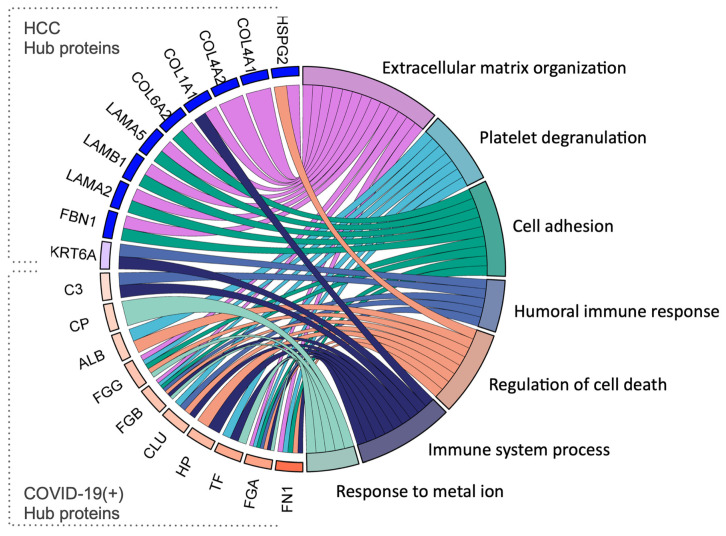
Functional interaction of the hub proteins. The chord diagram shows a detailed interaction of functional connections among the top 10 hub proteins (left side) from the HCC and COVID-19(+) groups and their assigned GO terms (right side).

**Figure 5 ijms-25-03704-f005:**
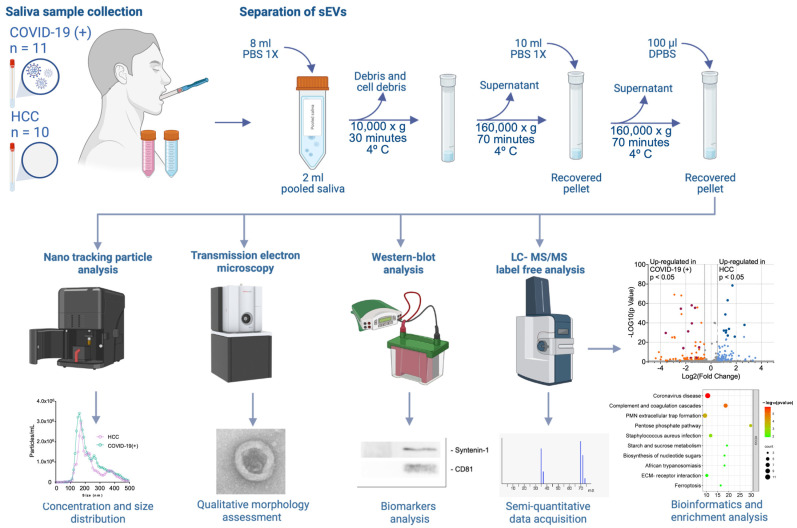
Schematic representation of differential ultracentrifugation-based sEV isolation and characterization methods in the context of symptomatic COVID-19 infection. Saliva samples were pooled based on clinical status (i.e., the COVID-19(+) and HCC groups). Differential ultracentrifugation-based extracellular vesicle isolation involved multiple cycles of centrifugation, ranging from 10,000× *g* up to 160,000× *g*. Following the final centrifugation, the supernatant was removed, and the enriched EV pellets were resuspended in 100 µL of DPBS. The concentration and distribution of sEVs were assessed using nano tracking particle analysis. Qualitative morphology assessment was conducted through transmission electron microscopy. EV markers (Syntenin-1, and CD81) and the contaminant marker (albumin) were evaluated by Western blot analysis. The proteomic cargo was determined by LC-MS/MS label-free analysis, and bioinformatics-assisted functional enrichment analyses were performed.

**Table 1 ijms-25-03704-t001:** Description of the demographic and clinical variables of the subjects included in the study.

	COVID-19(+) (*n* = 11)	HCC (*n* = 10)	Total (*n* = 21)
Gender	*p* = 0.0588		
Female	6 (54.5%) 5 (45.5%)	3 (30%) 7 (70%)	9 (42.86%) 12 (57.14%)
Male			
Age	31 (19)	54.5 (22)	40 (25)
Smoking	*p* = 0.4795		
No	8 (72.7%) 3 (27.3%)	8 (80%) 2 (20%)	16 (76.2%) 5 (23.8%)
Yes			
Comorbidities	*p* = 0.0000		
No	10 (90.9%) 1 (9.1%)	6 (60%) 4 (40%)	16 (76.2%)
Yes			5 (23.8%)
BMI	24.22 (7.25)	25.61 (9.47)	24.8 (6.33)

Data are presented through ‘frequencies (percentages)’ and ‘median (interquartile range)’. Abbreviations: BMI: body mass index; COVID-19(+): subject positive for COVID-19; HCC: Healthy Close Contact. A chi-squared test was conducted to compare groups.

**Table 2 ijms-25-03704-t002:** Top 10 hub proteins in the HCC and COVID-19(+) groups identified in CytoHubba using the MCC method.

Hub Proteins in the HCC Group			
Gene Name	Accession	Protein Name	Function (Uniprot)
*HSPG2*	P98160	Heparan sulfate proteoglycan core protein	An integral component of basement membranes. Responsible for the fixed negative electrostatic membrane charge. It serves as an attachment substrate for cells.
*COL4A1*	P02462	Collagen alpha-1(IV) chain	Type IV collagen is the major structural component of basement membranes, interacting with laminins and proteoglycans
*COL4A2*	P08572	Collagen alpha-2(IV) chain	Type IV collagen is the major structural component of basement membranes, interacting with laminins and proteoglycans
*COL1A1*	P02452	Collagen alpha-1(I) chain	Type I collagen is a member of the fibrillar-forming collagen.
*COL6A2*	P12110	Collagen alpha-2(VI) chain	Collagen VI acts as a cell-binding protein.
*LAMA5*	O15230	Laminin subunit alpha-5	Involved in attachment, migration, and organization of cells by interacting with other matrix components.
*LAMB1*	P07942	Laminin subunit beta-1	Involved in attachment, migration, and organization of cells by interacting with other matrix components.
*LAMA2*	P24043	Laminin subunit alpha-2	Involved in attachment, migration, and organization of cells by interacting with other matrix components.
*FBN1*	P35555	Fibrillin-1	Plays a key role in tissue homeostasis. Structural component of microfibrils of the extracellular matrix, which provide support to load-bearing connective tissues.
*KRT6A*	P02538	Keratin, type II cytoskeletal 6A	Involved in wound healing. Involved in the activation of follicular keratinocytes after wounding.
**Hub Proteins in the COVID-19(+) Group**			
**Gene Name**	**Accession**	**Protein Name**	**Function (Uniprot)**
*ALB*	P02768	Albumin	Binds water, Na+, K+, fatty acids, and hormones. Major zinc, calcium, and magnesium transporter in plasma.
*TF*	P02787	Serotransferrin	Transport of iron from sites of absorption and heme degradation to those of storage and utilization.
*CP*	P00450	Ceruloplasmin	Copper-binding glycoprotein with ferroxidase activity. It is involved in iron transport across the cell membrane.
*HP*	P00738	Haptoglobin	Captures free hemoglobin in plasma to allow hepatic recycling of heme iron and to prevent kidney damage. Also has antioxidant and antibacterial activity and plays a role in modulating the acute phase response.
*FGA*	P02671	Fibrinogen alpha chain	Together with FGB and FGG, polymerizes to form an insoluble fibrin matrix, acting as one of the primary components of blood clots.
*FGG*	P02679	Fibrinogen gamma chain	Together with FGA and FGB, polymerizes to form an insoluble fibrin matrix, acting as one of the primary components of blood clots.
*FGB*	P02675	Fibrinogen beta chain	Together with FGA and FGG, polymerizes to form an insoluble fibrin matrix, acting as one of the primary components of blood clots.
*CLU*	P10909	Clusterin	Prevents stress-induced aggregation of blood plasma non-native proteins.
*FN1*	P02751	Fibronectin	Involved in cell adhesion, cell motility, opsonization, wound healing, and maintenance of cell shape.
*C3*	P01024	Complement C3	C3 plays a central role in the activation of the complement system. It increases vascular permeability and causes histamine release.

## Data Availability

The proteomic data has been deposited on PRIDE [64], (https://www.proteomexchange.org/, accession number: PXD047446 on 1 December 2023). PRIDE is an official member of ProteomeXchange Consortium [65] which includes, PeptideAtlas [66], MassIVE [67], jPOST [68], iProX [69], and Panorama Public [70]. The Consortium was established to provide globally coordinated standard data submission and dissemination pipelines involving the main proteomics repositories and to encourage open data policies in the field. Deidentified clinical data can be made accessible upon publication, following approval of a proposal along with a signed data access agreement through the corresponding author. Only deidentified data that form the basis of the results presented in this article can be provided to investigators who have an approved proposal. Once a proposal is approved, clinical data can be shared via a secure online platform after the data access agreement has been signed.

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
