# Peer review of "Proteomic Analysis of Salivary Extracellular Vesicles from COVID-19 Patients Reveals a Specific Anti-COVID-19 Response Protein Signature"

_ijms, 2024, doi:10.3390/ijms25073704_

Round 1
Reviewer 1 Report
Comments and Suggestions for Authors
Weber and colleagues analyzed the proteomic profiles of saliva sEVs from COVID-19 (+) subjects and their healthy close contacts (HCC). The study design and data are of interest. The reviewer would suggest the following points for the authors to consider.
Major points:
1. Table 1: It appears that there is a marginal statistical difference between genders in two groups. Also, the difference in age between two groups seemed to be not minor. Could this contribute to the difference in sEVs? The authors shall discuss this.
2. Line 100 – 105: p value is not significant here. The authors shall revise the tone. This also applies to the later discussion regarding this set of data.
3. Line 136 - 158: In addition to the GO and pathway analysis of the identified proteins of each group individually, the authors shall use GO and pathway analysis to analyse the differentially expressed proteins between alone. This is to see if the differentially expressed proteins are COVID-related and shall be reflected in the main figure.
4. Line 376: the authors here described ‘pooled samples’. Please elaborate more regarding this. What samples were pooled? And what’s the rationale for pooling the samples?
5. Validation: in addition to proteomic analysis, the authors shall pick some of the top calls from the differentially expressed proteins for validation using western blot, flow cytometry, etc.
Minor points:
1. Figure 2b: for visualization purposes, the authors are encouraged to tag the names of the proteins (dots in the figure) that are most differentially expressed between groups, such as the top 10 differentially expressed ones.
2. Figure 3a and 3c: the protein names are too small to be seen. Please optimize the artwork so that the audience can read the names to make this figure meaningful.
3. Figure 5: if the authors used quantitative PCR to confirm COVID infection, the icon for rapid test shall be not used for illustration.
Comments on the Quality of English LanguageAcceptable
Author Response
We would like to thank the referees for the thorough review of our manuscript and their suggestions, which have improved the current version of our work. The point-by-point answers to the reviewers’ comments (in Italics) are as follows:
REVIEWER 1 (Comments to the Author):
Major points:
- Table 1: It appears that there is a marginal statistical difference between genders in two groups. Also, the difference in age between two groups seemed to be not minor. Could this contribute to the difference in sEVs? The authors shall discuss this.
Response: We agree with the referee’s observation regarding the potential influence of gender and age differences on the variations observed in salivary extracellular vesicles (sEVs) between COVID (+) patients and healthy controls. We have added a text in the manuscript, highlighted in yellow, discussing this issue.
- Line 100 – 105: p value is not significant here. The authors shall revise the tone. This also applies to the later discussion regarding this set of data –
Response: Thank you for your comment. A revised text has been added in the results and discussion sections. In addition, Figure 1 has been reorganized for a better description of the data.
- Line 136 - 158: In addition to the GO and pathway analysis of the identified proteins of each group individually, the authors shall use GO and pathway analysis to analyse the differentially expressed proteins between alone. This is to see if the differentially expressed proteins are COVID-related and shall be reflected in the main figure.
Response: Thank you for your valuable feedback. An additional enrichment analysis was conducted, and a figure was added as a supplementary file. Additionally, a description in the main text was added. However, it should be noted that enrichment analysis assesses a pool of features against a background. As many proteins are involved in different biological functions, their enrichment will depend mainly on two factors. One of them is the relationship between the set of features, and the other is related to how specific the proteins/features are to a certain term. To address this issue, an additional hierarchical cluster analysis has been added to evaluate the nature of the relationship of the enriched terms. As can be seen in Supplementary Figure 3, most of the enrichment terms identified in the individual group analysis (Figure 2) are represented in the whole assessment of differentially expressed proteins, regardless of their origin. Furthermore, the hierarchical cluster analysis reflects the grouping of enriched terms, highlighting that those enriched terms are closer together according to clinical status (i.e., Covid (+) and HCC).
- Line 376: the authors here described ‘pooled samples’. Please elaborate more regarding this. What samples were pooled? And what’s the rationale for pooling the samples?
Response: We appreciate the reviewer's comment. The revised version of the manuscript includes a rationale description to address this matter in both the methods and the discussion sections.
- Validation: in addition to proteomic analysis, the authors shall pick some of the top calls from the differentially expressed proteins for validation using western blot, flow cytometry, etc.
Response: We totally agree about the importance of validation studies in confirming the findings of proteomic analyses. However, it is important to note that our study was not designed for hypothesis-driven validation of specific protein targets. Instead, our study was designed as an exploratory investigation first to characterize the salivary extracellular vesicles from COVID-19 subjects compared to their healthy close contacts and second to analyze the proteomic cargo of these saliva extracellular vesicles, comparing coronavirus-diseased patients with healthy controls. In this context, our focus was on generating hypotheses and identifying potential candidates for further investigation in subsequent studies with a more targeted approach. We believe that the stringent experimental procedures—comprising the use of pooling samples, running blanks between samples, and conducting technical replicates—employed to ensure data quality, combined with the comprehensive proteomic analysis, provide valuable insights into the proteomic signature of salivary EVs in COVID-19. These findings lay the groundwork for future research efforts aimed at validating and further elucidating the functional significance of the identified proteins and their enriched-term annotations.
To clarify the stringent experimental procedures, a text was added in section “4.9. Liquid Chromatography–Tandem Mass Spectrometry (LC-MS/MS) analysis”.
Minor points:
- Figure 2b: for visualization purposes, the authors are encouraged to tag the names of the proteins (dots in the figure) that are most differentially expressed between groups, such as the top 10 differentially expressed ones.
Response: We have revised Figure 2b and enhanced the visualization of the differentially expressed proteins. We have incorporated the names of the proteins that are most differentially expressed between groups into the figure. Specifically, we have tagged the hub proteins in the COVID (+) group and the ten most representative proteins of the HCC group. It is worth noting that the hub proteins of the HCC group were mostly exclusive, with 9 out of 10 not represented in the volcano plot. By highlighting these key proteins, we aim to provide a clearer representation of the significant differences observed between the groups. We believe that these revisions improve the interpretability of Figure 2b and enhance the overall clarity of the manuscript.
- Figure 3a and 3c: the protein names are too small to be seen. Please optimize the artwork so that the audience can read the names to make this figure meaningful.
Response: We have taken into consideration the comment regarding the legibility of the protein names in Figures 3a and 3c. In response, we have optimized the artwork to ensure that the protein names are more visible, thereby enhancing the interpretability of the figures by including supplementary Figures 4 and 5, which offer a detailed view of the PPI network. Furthermore, we have updated the figure legend for Figure 3 to include a description addressing the content of these supplementary figures. These revisions aim to improve the clarity and effectiveness of our presentation.
- Figure 5: if the authors used quantitative PCR to confirm COVID infection, the icon for rapid test shall be not used for illustration.
Response: We have made the necessary revisions to Figure 5 to reflect the use of quantitative PCR (RT-qPCR) of nasopharyngeal swabs for confirming COVID infection. The icon has been updated accordingly to accurately depict the methodology employed in our research.
Reviewer 2 Report
Comments and Suggestions for Authors
Comments to the Authors
General comments:
The authors proposed a novel proteomics study to investigate EVs protein cargo in saliva COVID-19 patients. This work could be of high impact for the scientific literature because of the relevance of hot topics (COVID-19 and EVs). The study design is clear and proteomics data are well described, for example the chord diagram of hub proteins is very useful to show the significance of GO functions in COVID-19 patients and CTRL highlighting that salivary EVs of healthy volunteers trigger proteins involved in exocytosis and homeostatic mechanisms. However, many criticisms may arise, and, in my opinion, the limitations of the work are evident. For this reason, I think that the article is not suitable for publication in this form. Some specific comments for improving the manuscript are provided below as minor and major points.
Minor points
· The text contains many missing spaces before/after punctuation and units of measure. To revise the full article is recommended.
· Greater consistency of form between the various chapters is recommended.
· In “Material and Methods” section, the word “minute/minutes” is written in many ways. At the same time, the unit of measure “revolution per minute” are expressed in two different ways: “g” and “rpm”. To standardize the term for both measurements is recommended.
· In “Results” section, lines 102-115: the data described are not represented in the right panels of Figures 1. Maybe, these errors are typos of many corrections of the draft. I mean that panels G and H are absent in the figure legend. Lines 126: probably the authors are referring to Figure 2B and not 3B for Volcano Plot.
· Line 126: based on what is represented in the figure 2A, total proteins number “1427” is wrong. The total proteins number should be 1417, as reported in the abstract and in the Supplementary Table S1. At the same time, based on what is represented in the vend diagram and in the Volcano Plot, the number of up-regulated proteins in the HCC group is 72 instead of “71” as mentioned in the paper. In this regard, in the abstract 124 proteins are reported as differential, can the authors be clearer?
· Panels C-H of Figure 2: can the authors specify what is reported on the x-axis? Lines 144-145: the pathway “Steroid signaling pathway” is not mentioned in the figure 2D. Line 150: the molecular function “Extracellular constituent of cytoskeleton and epidermis’ is wrong. The authors probably added “extracellular” accidentally. Lines 157-158: biological processes “Leukocyte-mediated activation” and “PMN-mediated immunity” are not mentioned in the figure 2H, but they are representative of proteins up-regulated in COVID-19(+) patients.
· Line 182: the abbreviation for Salivary Extracellular Vesicles should be written without the S in capital letter.
Major points:
· Proteomics study is considered a first screening, and it is useful to understand EVs protein outfit. However, in this manuscript, any deeper functional validation of the data is missing. In my opinion, stronger functional and technical validations would be required, for example a western blot of differential proteins or an in vitro assay in order to make the diagnostic significance of their findings clearer.
· In general, after an accurate re-reading of the discussion, I suggest the authors to focus on the biological pathways of greatest interest for COVID-19 by selecting the most promising ones and thus perform other functional experiments. Overall, the idea behind this work is promising but the experiment showed in the manuscript are not strong enough to make any significant conclusion. In this regard, limitations of study are clear.
Author Response
We would like to thank the referees for the thorough review of our manuscript and their suggestions, which have improved the current version of our work. The point-by-point answers to the reviewers’ comments (in Italics) are as follows:
REVIEWER 2 (Comments to the Author):
General comments:
The authors proposed a novel proteomics study to investigate EVs protein cargo in saliva COVID-19 patients. This work could be of high impact for the scientific literature because of the relevance of hot topics (COVID-19 and EVs). The study design is clear and proteomics data are well described, for example the chord diagram of hub proteins is very useful to show the significance of GO functions in COVID-19 patients and CTRL highlighting that salivary EVs of healthy volunteers trigger proteins involved in exocytosis and homeostatic mechanisms. However, many criticisms may arise, and, in my opinion, the limitations of the work are evident. For this reason, I think that the article is not suitable for publication in this form. Some specific comments for improving the manuscript are provided below as minor and major points.
Minor points
- The text contains many missing spaces before/after punctuation and units of measure. To revise the full article is recommended.
Response: We appreciate the reviewer's feedback highlighting the need for improved spacing and punctuation throughout the text. We have revised the article to address these concerns.
- Greater consistency of form between the various chapters is recommended.
Response: We have considered this feedback and have made the necessary adjustments to ensure uniformity throughout the manuscript.
- In “Material and Methods” section, the word “minute/minutes” is written in many ways. At the same time, the unit of measure “revolution per minute” are expressed in two different ways: “g” and “rpm”. To standardize the term for both measurements is recommended.
Response: We have carefully reviewed and standardized the terminology for "minute/minutes". However, only one term referring to RPM was left, as this term was associated with a distinctive procedure in the Mass spectrometry workflow (“…cleaned peptides were dried in a rotary concentrator at 1000 rpm overnight…”). Additionally, while revolutions per minute (RPM) can be transformed into relative centrifugal force (RCF) or g force (x g) using the formula RCF = (RPM)^2 × 1.118 × 10^-5 × r, we did not specify the exact value of 'r' for radius.
- In “Results” section, lines 102-115: the data described are not represented in the right panels of Figures 1. Maybe, these errors are typos of many corrections of the draft. I mean that panels G and H are absent in the figure legend. Lines 126: probably the authors are referring to Figure 2B and not 3B for Volcano Plot.
Response: We greatly appreciate the reviewer's feedback. We have thoroughly revised the article to address these concerns.
- Line 126: based on what is represented in the figure 2A, total proteins number “1427” is wrong. The total proteins number should be 1417, as reported in the abstract and in the Supplementary Table S1. At the same time, based on what is represented in the vend diagram and in the Volcano Plot, the number of up-regulated proteins in the HCC group is 72 instead of “71” as mentioned in the paper. In this regard, in the abstract 124 proteins are reported as differential, can the authors be clearer?
Response: We appreciate the reviewer's feedback. We have thoroughly revised the article to address these concerns. Additionally, text was added to the legend of Figure 2, detailing the 124 differentially expressed proteins found in the COVID-19 (+) group.
- Panels C-H of Figure 2: can the authors specify what is reported on the x-axis? Lines 144-145: the pathway “Steroid signaling pathway” is not mentioned in the figure 2D. Line 150: the molecular function “Extracellular constituent of cytoskeleton and epidermis’ is wrong. The authors probably added “extracellular” accidentally. Lines 157-158: biological processes “Leukocyte-mediated activation” and “PMN-mediated immunity” are not mentioned in the figure 2H, but they are representative of proteins up-regulated in COVID-19(+) patients.
Response: We greatly appreciate the reviewer's feedback. We have thoroughly revised the Figure 2 and the text in the mentioned lines to address these concerns.
- Line 182: the abbreviation for Salivary Extracellular Vesicles should be written without the S in capital letter.
Response: Done.
Major points:
- Proteomics study is considered a first screening, and it is useful to understand EVs protein outfit. However, in this manuscript, any deeper functional validation of the data is missing. In my opinion, stronger functional and technical validations would be required, for example a western blot of differential proteins or an in vitro assay in order to make the diagnostic significance of their findings clearer.
Response: We recognize the importance of validation in strengthening the diagnostic significance of our study. Our research was primarily exploratory, focusing on to separate and characterize salivary EVs (size, concentration, morphology, surface markers) from subjects with COVID-19 (+) and their healthy close contacts (HCC) and to explore their proteomic cargo by a mass-spectrometry-based proteomics approach. While we acknowledge the importance of validation studies in confirming the findings of proteomic analyses, it is important to note that the evidence related with salivary EVs in COVID-19 is scarce and the result of this exploratory study contributes to the characterization of salivary EVs in COVID-19 infection. Additionally, our study was not designed for hypothesis-driven validation of specific protein or functional targets. Instead, our focus was on generating hypotheses and identifying potential candidates for further investigation in subsequent studies with a more targeted approach. To the best of our knowledge, this is the first study characterizing salivary EVs and their proteomic cargo from symptomatic COVID-19 diseased patients compared to their healthy close contact controls.
We believe that the stringent experimental procedures — comprising the use of pooling samples, running blanks between samples, and conducting technical replicates — employed to ensure data quality, combined with the comprehensive proteomic analysis, provide valuable insights into the proteomic signature of salivary EVs in the context of COVID-19. To clarify the stringent experimental procedures, a text was added in section “4.9. Liquid Chromatography–Tandem Mass Spectrometry (LC-MS/MS) analysis”. Additionally, a discussion on the limitations of the present study was added.
Nevertheless, we believe that the findings presented in our study provide a foundation for future research efforts aimed at validating and further elucidating the functional significance of the identified proteins and their enriched-term annotations.
- In general, after an accurate re-reading of the discussion, I suggest the authors to focus on the biological pathways of greatest interest for COVID-19 by selecting the most promising ones and thus perform other functional experiments. Overall, the idea behind this work is promising but the experiment showed in the manuscript are not strong enough to make any significant conclusion. In this regard, limitations of study are clear.
Response: We greatly appreciate the reviewer's thorough evaluation of our discussion section and their insightful suggestion regarding the focus on biological pathways most pertinent to COVID-19. While we acknowledge the potential benefits of conducting additional functional experiments to further elucidate the biological pathways highlighted in our study, we regret to inform you that due to funding constraints, we are currently unable to perform such validations or additional experiments. However, we agree that the limitations of our study are evident, and we have duly emphasized them in the manuscript. Despite these constraints, we firmly believe that our work offers valuable preliminary insights into the proteomic signature of salivary extracellular vesicles in symptomatic COVID-19 patients.
We are optimistic that our findings will stimulate further research projects in this area, including more comprehensive studies incorporating functional validations. In fact, we are actively seeking additional funding to support the continuation of this research line. We sincerely thank the reviewer for their constructive feedback and want to assure them that we have taken note of their suggestion for future research directions.
Round 2
Reviewer 2 Report
Comments and Suggestions for Authors
As highlighted in the first revision round, the authors deal with many of the hottest scientific topics (i.e., EVs, COVID-19, proteomics…), for this reason, it could be of high impact especially due to the in vivo study design. I mean that the authors used the saliva of patients to isolate salivary EVs and they didn’t manage EVs isolated from cell cultures. As a matter of fact, when we want to characterize EV protein cargo from a biological fluid, the most challenging problem is the isolation step that must be suitable for proteomics analysis by LC-MS/MS. In this regard, patients’ variability is evident and pooling sample is necessary due to this constraint: this could represent a strong limitation for the study as well as sample size. I perfectly know the problems related to EV management for proteomics purposes when working with biofluid. The authors have made substantial changes to this article in response to the reviewer’s comments, including some additional information and results by revising proteomics results in terms of Volcano Plot and number of quantified proteins. Now, it sounds much better in understanding and fluency of the whole text. Even if the authors are unable to perform other validation experiments, this study sounds like an exploratory work on salivary EVs from COVID-19 patients. So, as the study design is relevant and clear and their findings are now present as preliminary results in line with COVID-19 and its clinical picture, I strongly believe that this manuscript could stimulate further research in its application field. As a matter of fact, all potential limitations of the study in terms of functional validations and sample size are now highlighted in the discussion section. In my opinion, the manuscript can be published in this form according to the guidelines of the journal. In conclusion, I want to encourage the authors to validate their results and continue this research line.
Author Response
We sincerely appreciate the referee's comments on the revised version of our manuscript. As suggested, we will pursue this research line using appropriate methodologies to validate our results.